# The Relationship between Penetration, Tension, and Torsion for the Fracture of Surimi Gels: Application of Digital Image Correlation (DIC)

**Hyeon Woo Park** [1,2], **Jae W. Park** [2] and **Won Byong Yoon** [1,3,*]

[1] Department of Food Science and Biotechnology, College of Agricultural and Life Science, Kangwon National University, Chuncehon 24341, Republic of Korea
[2] OSU Seafood Research and Education Center, Oregon State University, Astoria, OR 97103, USA
[3] Elderly-Friendly Food Research Center, Agriculture and Life Science Research Institute, Kangwon National University, Chuncheon 24341, Republic of Korea
[*] Correspondence: wbyoon@kangwon.ac.kr

**Abstract:** A standardized method to evaluate the material properties of surimi gels has to be updated because of the lack of accuracy and the repeatability of data obtained from conventional ways. To investigate the relationships between the different texture measurement methods used in surimi gels, 250 batches of different surimi gels were used. The textural properties of surimi gels made with or without whey protein concentrate (SG-WP), potato starch (SG-PS), or dried egg white (SG-EW) were measured under torsion, tensile, and penetration tests. The correlation between the textural properties related to the deformation and hardness of surimi gels without any added ingredients (SG) was linear ($R^2 > 0.85$). However, the $R^2$ values of the shear strain and tensile strain of SG-WP and SG-EW were significantly lower than that of SG. The strain distributions of surimi gels with and without added ingredients were estimated by digital image correlation (DIC) analysis. The results showed that the local strain concentration in SG-WP and SG-EW was significantly higher than that of SG in the failure ring tensile test and the torsion test ($p < 0.05$). DIC analysis was an effective tool for evaluating the strain distribution characteristics of surimi gels upon fracture from torsion, penetration, and tension.

**Keywords:** surimi; fish protein; penetration test; punch test; ring tensile test; torsion test; image analysis; local strain





## 1. Introduction

Surimi, stabilized fish myofibrillar protein, is a major ingredient of surimi seafood products. Surimi seafood products have become increasingly popular due to their unique textural properties as well as their high nutritional value [1]. With suitable thermal treatment, surimi becomes a highly deformable gel. The textural properties of surimi seafood products are mainly characterized by elastic and/or viscoelastic properties [2,3]. Several mechanical tests are used to characterize the textural properties of surimi seafood, such as the penetration test, torsion test, and ring tensile test [2,4,5]. The penetration test has been widely used in the surimi industry because of the simplicity of the measuring procedure [4,6–8]. The torsion test is a unique measurement that provides failure shear stress and failure shear strain, with a high correlation with sensory results [1]. Ring tensile tests with image analysis have also been used to measure the failure tensile properties of surimi gels [5]. Measurements from the above three methods may reflect the different textural characteristics of food gels. Thus, a more comprehensive understanding of surimi gel properties could be obtained by comparing measurements from all three methods.

The digital image correlation method (DIC) is an emerging non-contact optical technology for measuring displacement and strain. DIC works by comparing the digital images of

a component or test specimen at different strains. By tracking a random speckle pattern, the system can measure displacement and strain on the surface. Since this technique does not need a complicated monitoring system, the measurement has been widely used in the fields of fracture mechanics, wood products, food products, and inverse stress analysis [9–11]. Additionally, compared to other methods that utilize the interference of light waves, the phase analysis of the fringe pattern and subsequent phase unwrapping process are not necessary. Another important advantage of the DIC technique compared to mechanical tests is that it can identify local strain during the mechanical tests while the mechanical tests only measure the overall strain. Therefore, the DIC technique can provide additional information to compare three mechanical tests, such as the penetration, torsion, and ring tensile tests.

The structural development of food gels depends on the structure of individual components and the interactions between the components in the mixture. The textural properties of surimi seafood are highly associated with the structure and interactions of the ingredients in the surimi seafood formulation. Most surimi seafood contains egg whites, whey protein, or potato starch to control the texture and cost of the final product [12–15]. The contribution of the ingredients to the textural properties of surimi gels is evaluated by measuring fundamental mechanical properties by compressive, tensile, or torsion tests [1]. As a homogenous isotropic material, for example, surimi gels are fundamentally tested, and the mechanical properties are solely dependent upon the direction of the load applied for each fundamental mechanical test. Numerous studies have characterized the effect of ingredients on textural properties by measuring mechanical properties [16–19]. However, there are no reports of studies evaluating the effect of ingredients on the texture properties of surimi gels using different fundamental mechanical tests. Therefore, a better understanding of the effects of the ingredients on penetration, failure ring tensile, and torsion tests for surimi and surimi seafood is needed.

The objectives of this study were to (1) characterize the relationship between the penetration test, failure ring tensile test, and torsion test, (2) identify the effects of egg whites, whey protein, and potato starch on the texture characteristics of surimi gel, and (3) analyze the different relationships associated with the ingredients using the digital image correlation method.

## 2. Materials and Methods

### 2.1. Materials

A total of 250 batches of surimi with and without ingredients with different moisture contents were used. Alaska pollock grades SA, KA, and B were purchased from American Seafoods Co. (Seattle, WA, USA), grades A and KA were purchased from Unisea Inc. (Seattle, WA, USA), grade A was purchased from Trident Seafoods Co. (Seattle, WA, USA), grade A was purchased from Arctic Storm Inc. (Seattle, WA, USA), and grade FA was purchased from Alaska Ocean Seafood (Anacortes, WA, USA). Pacific whiting grades AA and A were purchased from Ilwaco Fish Co. (Ilwaco, WA, USA) and American Seafoods Co. (Seattle, WA, USA). Surimi was cut into about 1 kg blocks, vacuum-packaged, and stored in a freezer ($-25\,^\circ$C) throughout the experiments. The initial moisture content of each surimi lot was measured according to the AOAC method [20]. Dried egg whites (Henningsen Foods Inc., Omaha, NE, USA), whey protein concentrate (8200, Hilmar Ingredients, Hilmer, CA, USA), and potato starch (Emsland Group, Emlichheim, Germany) were used as ingredients in the surimi gels.

### 2.2. Surimi Gel Preparation

Surimi paste and gel were prepared according to the method of Park et al. [5]. After thawing at 25 $^\circ$C for 1 h, surimi was cut into 5 cm cubes. A Stephan vacuum cutter UM-5 (Stephan Machinery Corp., Columbus, OH, USA) was used to make surimi paste. In the first 1 min, frozen cubes were chopped at low speed. Salt (20 g/kg) was sprinkled in, and chopping was continued at low speed for 1 min. Ice/water (0 $^\circ$C) with or without ingredients (0–90 g/kg), such as dried egg whites (SG-EW), whey protein concentrate (SG-

WP), and potato starch (SG-PS), was added to adjust the moisture level to 740–810 g/kg, and the samples were chopped at low speed for 1 min. For the final 3 min, chopping was continued at high speed while a vacuum was maintained at 0.4–0.6 bar. During chopping, a constant cold temperature (<8 °C) was maintained using a NesLab chiller (NesLab, Portsmouth, NH, USA). The surimi paste was stuffed into stainless steel tubes (length, 17.5 cm and inner diameter, 3.0 cm) and dumbbell-shaped tubes (maximum diameter, 19 mm and minimum diameter, 10 mm) using a sausage stuffer (Sausage Maker, Buffalo, NY, USA). The interior wall of the tubes was coated with a film of PAM cooking spray (Boyle-Midway, Inc., New York, NY, USA). Then, the tubes were heated in a water bath at 90 °C for 30 min. The cooked gels were chilled quickly in ice water (0 °C). The gels were kept refrigerated (4 °C) overnight.

*2.3. Texture Analysis*

2.3.1. Penetration Test

Penetration tests were performed with a TA-XT texture analyzer (Stable Micro Systems, Surrey, UK) equipped with a spherical plunger (1 mm/sec of crosshead speed and 5 mm of diameter). Surimi gels at 4 °C were placed at 25 °C for 3 h prior to gel testing. Cylinder-shaped surimi gels (25 mm long) were prepared and subjected to fracture by penetration. The penetration distance and breaking force were measured in mm and g, respectively. All tests were conducted 10 times.

2.3.2. Ring Tensile Test

A failure ring tensile test was performed using a TA-XT texture analyzer equipped with 2 pins (diameter 10 mm) according to the method of Park and Yoon [5]. The cylindrical gels were cut into disk shapes (diameter, 30 mm; length, 10 mm). The ring-shaped samples were prepared by perforating the gels with a ring cutter. The ring sample dimensions had a width of 10 mm, an inside diameter of 17 mm, and a thickness of 5 mm. The pin at the top of the equipment was moved up to fracture the ring-shaped samples by tension (constant speed of 1 mm/sec). All experiments were conducted 10 times.

In the ring tensile test in this study, failure hoop stress and failure uniaxial tensile stress were estimated to determine gel hardness (failure ring tensile stress). Failure hoop stress relates the wall circumferential stress to the internal pressure and the geometry of the ring specimen. Failure hoop stress was calculated by the following equation [5]:

$$\sigma_h = \frac{F}{wD} \tag{1}$$

where $\sigma_h$ is the failure hoop stress, $F$ is the load measured during ring tensile testing, $w$ is the width of the ring specimen, and $D$ is the inside diameter of the ring specimen. In failure uniaxial tensile stress, the sample is subjected to tension by opposing forces along its axis. Failure uniaxial tensile stress is calculated based on the cross-sectional area of the specimen as follows:

$$\sigma_u = \frac{F}{2wT} \tag{2}$$

where $\sigma_u$ is the failure uniaxial tensile stress, and $T$ is the thickness of the ring specimen.

In ring tensile testing, the ring specimen is deformed in the circumferential direction. Failure ring tensile strain was estimated according to Dieter [21], based on the dimensionless true strain by using the following equation:

$$\varepsilon = ln\frac{C}{C_0} \tag{3}$$

where $\varepsilon$ is the failure ring tensile strain, $C$ is the inside circumference of ring specimen, and $C_0$ is the initial inside circumference of the ring specimen.

### 2.3.3. Torsion Test

A torsion test was performed with a torsion gelometer (Gel Consultants, Raleigh, NC, USA). Cold gels (4 °C) were placed at room temperature for 2 h to equilibrate gel temperatures before testing. Dumbbell-shaped samples were measured in the gelometer for torsional shear at a rotational rate of 2.5 rpm. All experiments were conducted 10 times.

Shear stress and shear strain at mechanical failure were measured. Shear stress and shear strain indicate gel hardness and deformability of the gel, respectively [22,23]. Failure shear stress occurs at the boundary of the minimum cross-section for this specimen geometry. The shear stress value was calculated by the following equation [24,25]:

$$\tau = \frac{M_t r_{min} K}{J} \qquad (4)$$

where $\tau_{max}$ is the shear stress, $M_t$ is the torque, $r_{min}$ is the radius of the smallest cross-section of the torsion specimen, $K$ is the shape factor constant, and $J$ is the polar moment of inertia of the smallest cross-section.

The shear strain is defined as follows [26]:

$$\psi_{total} = \psi_Q + \psi_U = Q\frac{M_t}{G} + U\frac{M_t}{G} \qquad (5)$$

$$\psi_Q = \psi_{total}\frac{\psi_Q}{\psi_Q + \psi_U} = \psi_{total}\frac{Q}{Q + U} \qquad (6)$$

$$\gamma = \frac{2K\psi_Q}{\pi r_{min}^3 Q} \qquad (7)$$

where $\psi_{total}$ is the total angular rotation, $\psi_Q$ is the angular rotation in the $Q$ section, $\psi_U$ is the angular rotation in the $U$ section, $Q$ is the specimen shape factor of the $Q$ section, $U$ is the specimen shape factor of the $U$ section, $G$ is the shear modulus, and $\gamma$ is the dimensionless shear strain (Figure 1). Shear strain at the failure point was evaluated based on the true strain defined according to Nadai [25] using the following equation:

$$\gamma_t = ln\left[1 + \frac{\gamma^2}{2} + \gamma\left(1 + \frac{\gamma^2}{4}\right)^{1/2}\right] \qquad (8)$$

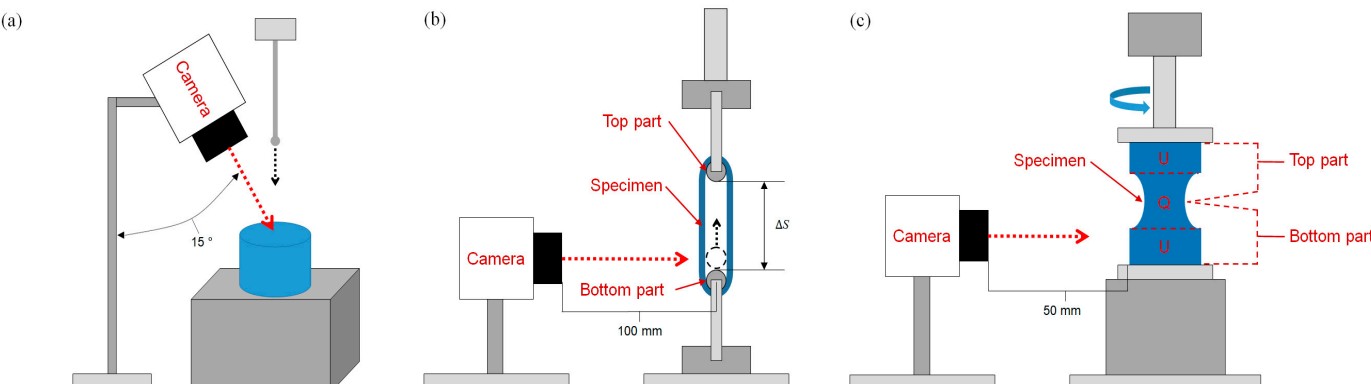

**Figure 1.** The image acquisition systems for the (**a**) penetration test, (**b**) ring tensile test, and (**c**) torsion test.

The material it is based on is elastic. Since the shear strain at failure is desired, if a material does not behave elastically up to the point of failure, difficulty arises in obtaining the actual failure shear strain value [21].

*2.4. Digital Image Correlation (DIC)*

Digital image correlation is a versatile, non-contact optical technique for displacement and deformation measurements, and this technique has been used in experimental mechanics to understand strain fields and whole-field displacement [27]. It compares random speckle patterns on the surface of specimens. The images of the deformed specimen are compared to the original speckle pattern of the undeformed specimen. Displacement and local strain are calculated using a pattern-matching technique. In this study, the speckle patterns of surimi gel were created by the manual spraying of black ink (Pelikan 4001, Pelikan Inc., Schindellegi, Switzerland). Each image was divided into different subsets containing multiple pixels for tracing since it was not practical to compare each pixel in the image.

Pattern matching is based on obtaining a correlation between the subsets of images in the deformed and undeformed states. The monitored successive frames in this study were used for DIC analysis in MATLAB (Mathworks ® Inc., Natick, MA, USA). A full description of this DIC method used to estimate the strain distribution can be found in Jones et al. [28].

2.4.1. Image Acquisition

The image acquisition systems for the penetration test, failure ring tensile test, and torsion test were developed as shown in Figure 1. A digital compact camera (COOLPIX S6100, Nikon Co., Tokyo, Japan) with a lens (NIKKOR 5–35 mm $f$/3.7–5.6 25° FOV, Nikon Co., Tokyo, Japan) was used to record each test at frame rates of up to 30 fps with a resolution of 0.95 million pixels during each test. For the penetration test, the camera was located to record the penetration point of the surimi gel at an angle of 15° (Figure 1a). To record the front view of the sample in the failure ring tensile test, the camera was placed horizontally over the sample at a distance of 140 mm (Figure 1b). For the torsion test, the camera was placed horizontally over the sample to focus on the center of the bottom part at a distance of 70 mm (Figure 1c). Images were obtained in PNG format from the video frames at each time interval. The obtained successive images were transferred to the computer for automated image processing followed by DIC analysis. The white objects (surimi gels) placed on a contrasting background in the successive images were extracted by a threshold-based segmentation algorithm [5]. The processed images were used for DIC analysis.

2.4.2. Calculation of Local Strain

For the penetration test, a horizontal, rectangular area of the surimi gel (vertical length = 100 pixels) was used to calculate local strain, which had a penetration point at the center. Failure ring tensile strain was calculated using the front side of the ring specimen. To calculate the local strain of the torsion test, the displacement of the bottom part was used, in which the top part was rotated (Figure 1c). To evaluate local strain, the local strain concentration degree was developed based on the deviation concept using the following equation:

$$D_c = \frac{1}{N} \sum_{i=1}^{N} \left[ \frac{100}{S_{max}} (S_{max} - S_i) \right] \tag{9}$$

where $D_c$ is the local strain concentration degree, $S_i$ is the average local strain of the subset line, and $S_{max}$ is the maximum value of $S_i$.

*2.5. Statistical Analysis*

All experiments were conducted 10 times. Mean values and standard deviations were determined. One-way ANOVA and Tukey's multiple comparison tests were carried out at the significance level of 0.05 by using SPSS software v27 (SPSS Inc, Chicago, IL, USA).

**3. Results and Discussion**

*3.1. Textural Properties Related to the Hardness of Surimi Gel*

The characteristics of the textural properties of surimi gel were identified in 250 batches of surimi gel. Failure hoop stress and failure uniaxial tensile stress were estimated to

examine the correlation between different textural properties related to the hardness of surimi gels, as shown in Figure 2. It should be noted here that breaking force, shear stress, and failure hoop stress showed linearity regardless of the added ingredient ($R^2 > 0.90$), whereas the correlation between failure uniaxial tensile stress and breaking force and shear stress did not show linearity or any other trends ($R^2 < 0.73$). In the ring tensile tests, uniaxial tensile stress is used to estimate the hardness of ductile materials, such as alloy 690 and Zircaloy [29,30], while hoop stress is used for elastic materials, such as surimi gel and tissue [5,31]. Many studies reported a linear relationship between the hardness of food gels measured by different methods. According to Chung and Lee [32], the compressive force, penetration force, and tensile force of various surimi gel products tended to increase proportionally. Park [1] also reported that the correlation between breaking force and shear stress of surimi gel showed a linear relationship. Similar results were also observed in texture related to the hardness of food gels [33]. Therefore, the results indicated that the hardness of surimi gel in the failure ring tensile test should be estimated based on failure hoop stress.

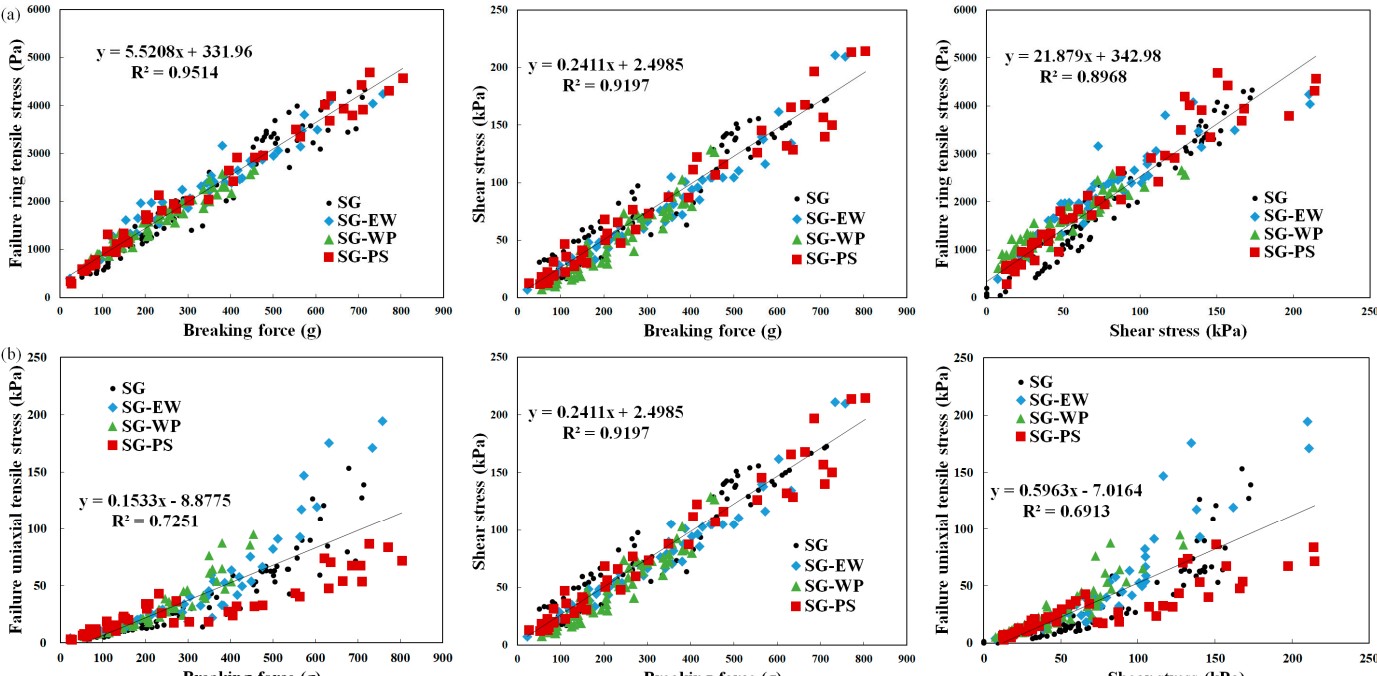

**Figure 2.** The correlation of textural properties related to the hardness of surimi gels. (**a**) Failure hoop stress and (**b**) failure uniaxial tensile stress. The trend line for each figure was based on surimi gel without any added ingredients.

### 3.2. Textural Properties Related to Hardness of Surimi Gels

As shown in Figure 3, the correlation between different textural properties related to the deformation of SG showed a linear function ($R^2 > 0.85$) regardless of the penetration distance, failure ring tensile strain, or shear strain. The correlations of SG-PS showed tendencies similar to those of SG. However, a non-linear correlation between SG-EW and SG-WP was observed when the penetration distance was compared to failure ring tensile strain and shear strain. While potato starch granules fill the interstitial spaces of the fish protein network and swell in the water surrounding the protein matrix [1,34], protein interactions between surimi and protein additives, such as egg whites and whey protein, occur during gelation in mixed protein systems, affecting gel structure and texture [35,36]. Such interactions might affect the failure point of the tests, which are dependent upon the matrix material and loading direction [35–37]. Hamann et al. [33] reported that strain, as an indicator of protein interactions, was strongly affected by protein functionality. Truong

and Daubert [38], who compared large strain methods in various food gels, reported that the correlation between compression, torsion, and vane tests was different in gellan gel, calcium-sulfate-coagulated tofu, and silken tofu. Truong and Daubert [39] also reported that the relationship between the vane and the torsion tests differed in various kinds of cheese. Penetration distance increased as shear strain and failure ring tensile strain increased, and then penetration distance converged at about 14 mm while shear strain and failure ring tensile strain increased. When very highly deformable gels are tested, the penetration distance might be influenced by the extremely compressed gel matrix, resulting in a converged penetration distance value in contrast to failure ring tensile strain and shear strain because the magnitude of the displacement in the penetration test is limited by the height of the specimen in contrast to the failure ring tensile test and torsion test [5,33]. In this study, 25 mm high surimi gels with a spherical plunger (diameter 5 mm) were used for the penetration test.

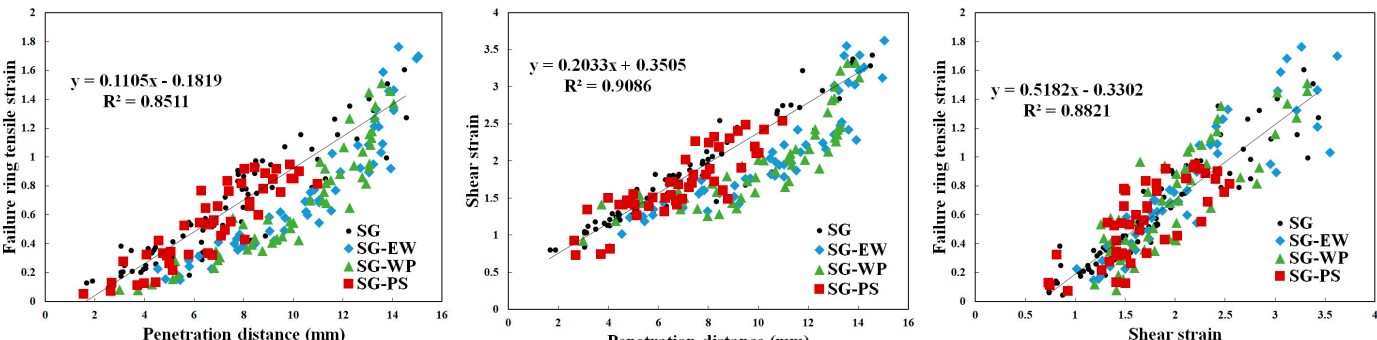

**Figure 3.** The correlation of textural properties related to the deformation of surimi gels. The trend line for each figure was based on surimi gel without any added ingredients.

### 3.3. Evaluation of Local Strain

Based on the relationship between textural properties related to deformation (Figure 3), Alaska pollock surimi gels (A grade), which showed a similar penetration range, were chosen as representative surimi gels to compare the strain properties (Table 1). Surimi gels were made without (AP) and with 3% dried egg white, whey protein concentrate, or potato starch (AP-EW, AP-WP, and AP-PS, respectively). Failure ring tensile strain and shear strain of different surimi gels with or without added ingredients were significantly different in contrast to the penetration distance results, which showed similar value ranges. The strain values of AP-EW and AP-WP were lower than that of AP in the failure ring tensile test and torsion test. These differences related to local strain could be explained by the different effects of the ingredients in the formation of the network structure of the surimi gels [1,35–37,40].

The local strain contour plot of the surimi gels in the penetration test, failure ring tensile test, and torsion test was estimated by DIC analysis (Figure 4). DIC analysis of surimi specimens was conducted until failure of the surimi gels, and the picture was taken at the failure point of each surimi sample. The failure ring tensile test and the torsion test results for AP and AP-PS showed that local strain for the entire region increased compared to the AP-EW and AP-WP results, in contrast to the punch test results, where local strain was concentrated on the failure point regardless of the ingredients. The maximum local strain values in the punch test, failure ring tensile test, and torsion test were not significantly different ($p > 0.05$), at 94.9% ($\pm$0.59), 132.5% ($\pm$5.3), and 74.2% ($\pm$1.8), respectively. In contrast, textural property results related to the deformation of the gels were significantly different ($p < 0.05$) (Table 1). Therefore, the differences in the relationship between penetration, tension, and torsion could be explained by the local strain concentration.

**Table 1.** Strain properties of the surimi gels.

| Strain Property | Test Method | Surimi Gel | | | |
|---|---|---|---|---|---|
| | | AP | AP-EW | AP-WP | AP-PS |
| Textural properties | Penetration test (mm) | 7.12 ± 0.04 a* | 6.97 ± 0.16 ab | 6.86 ± 0.21 b | 6.94 ± 0.09 b |
| | Failure ring tensile test | 0.53 ± 0.04 b | 0.41 ± 0.05 c | 0.35 ± 0.04 c | 0.66 ± 0.07 a |
| | Torsion test | 1.58 ± 0.06 a | 1.37 ± 0.09 bc | 1.35 ± 0.08 c | 1.49 ± 0.06 b |
| Maximum local strain (%) | Penetration test | 95.3 ± 3.8 a | 94.2 ± 3.3 a | 95.4 ± 2.5 a | 94.5 ± 2.2 a |
| | Failure ring tensile test | 132.9 ± 7.7 a | 130.0 ± 9.2 a | 127.4 ± 14.2 a | 139.6 ± 10.5 a |
| | Torsion test | 76.5 ± 5.8 a | 74.7 ± 11.8 a | 72.8 ± 11.6 a | 72.6 ± 5.4 a |
| Local strain concentration (%) | Penetration test | 87.1 ± 0.5 a | 86.9 ± 0.8 a | 86.8 ± 1.0 a | 88.2 ± 1.0 a |
| | Failure ring tensile test | 26.2 ± 5.0 bc | 35.1 ± 3.0 a | 36.5 ± 5.5 ab | 24.2 ± 3.2 c |
| | Torsion test | 31.3 ± 6.4 b | 51.2 ± 4.9 a | 49.8 ± 3.5 a | 32.2 ± 4.7 b |

* Different letters in the same row indicate that values are significantly different ($p < 0.05$).

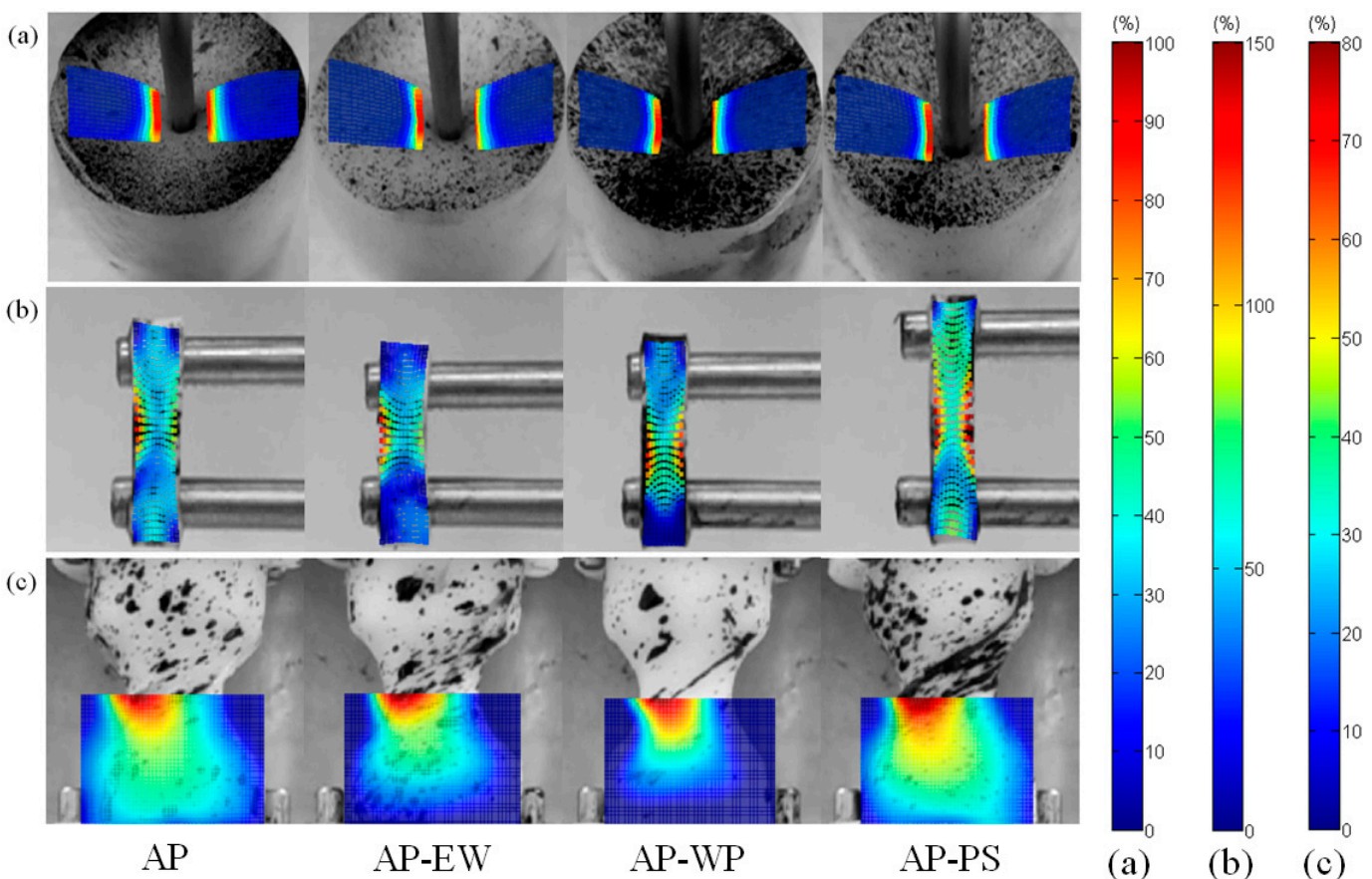

**Figure 4.** Representative contour plots of local strain components for the (**a**) penetration test, (**b**) failure ring tensile test, and (**c**) torsion test at failure points in each surimi gel.

To quantify the local strain of the surimi gels in the penetration, tensile, and torsion tests, the local strain concentration was evaluated according to Equation (9) (Table 1). As expected, although significant differences in the failure ring tensile test and torsion test were observed, there was no significant difference in the degree of local strain concentration in the penetration test ($p > 0.05$). Fracture strain is determined by deforming a sample to the point of abrupt mechanical yield [33]. Consequently, concentrated local strain can result in the early fracture of gels. Highly elastic materials showed no significant difference in local strain in the tensile test [5]. Protein interactions between surimi and protein additives, such as egg whites and whey protein, occur during gelation in mixed protein

systems [36]. Chaijan et al. [41] and Panpipat et al. [42] reported that increases in protein–protein interactions decreased water–protein interactions, thereby resulting in a decrease in elasticity. Wasinnitiwong et al. [18] reported that egg whites reduced elasticity in surimi gels. Hastings and Currall [43] also reported that the effect of egg whites was to reduce elasticity in cod surimi gel in contrast to potato starch, which increased the elasticity of surimi gel. Starch granules act as fillers in the gel matrix [44]. Zhou et al. [45] reported that the addition of unmodified egg white reduced the elasticity of surimi gels. The granules produce reinforcement in the gel matrix after they absorb water and swell without protein interaction [46]. Park and Yoon [5] reported that surimi gels without any added ingredients showed similar local strain values for the entire region in the failure ring tensile test. The average local strain concentration of 250 different surimi gels was also evaluated (Figure 5). The results were similar (Table 1). The local strain concentration of SG-EW and SG-WP was lower than that of SG, which made a difference in the correlation between penetration, torsion, and ring tensile tests. The difference increased as textural properties related to the deformation of surimi gel increased, and then, as mentioned above, the difference decreased as the penetration distance was increased to the limit, which was about 14 mm. Thus, our observations demonstrated that SG-EW and SG-WP showed higher local strain concentration in the failure ring tensile and torsion tests, which could result in the early fracture of surimi gels during testing. While the penetration test induced concentrated local strain at the fracture point of surimi gels regardless of the ingredients, the ring tensile test and torsion test showed different strain distributions of the surimi gels by the addition of egg whites and whey protein, which might cause changes in elasticity due to the interaction between the different proteins. Local strain analysis using the DIC technique could help in understanding the characteristics of the different texture measurement methods [47–49].

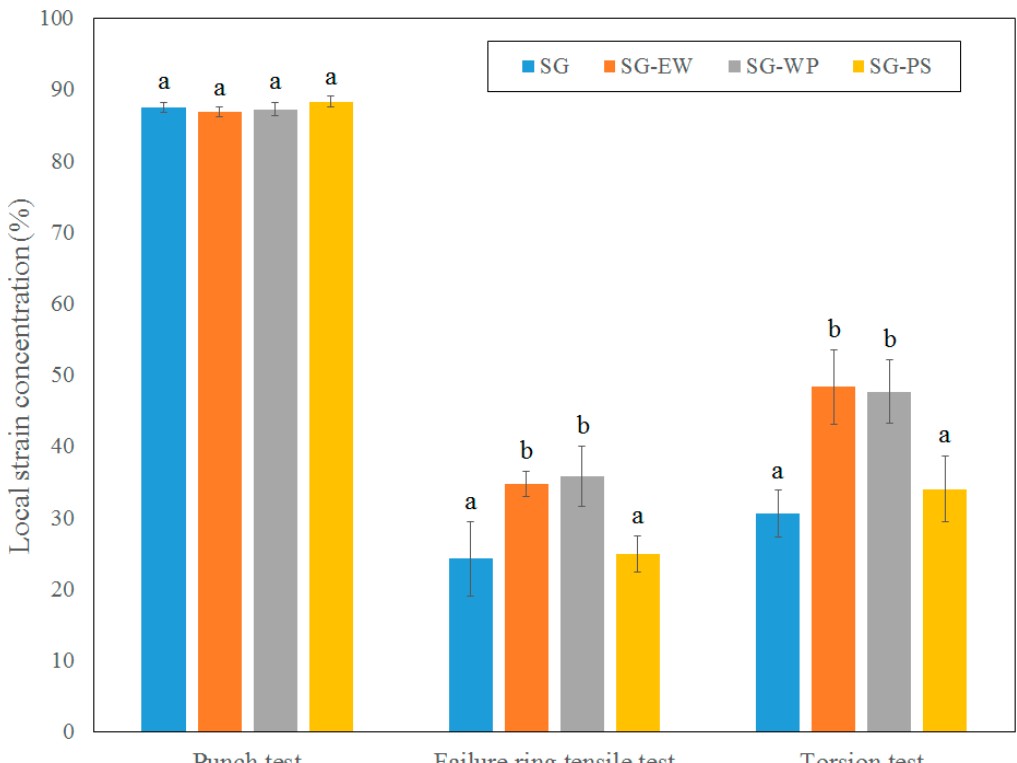

**Figure 5.** Local strain concentrations for the penetration test, failure ring tensile test, and torsion test. Different letters in each test method indicate that values are significantly different ($p < 0.05$).

## 4. Conclusions

This study investigated the fracture properties of surimi gels with and without added ingredients. The textural properties related to the hardness and the deformation of surimi

gels were measured by penetration, ring tensile, and torsion tests, which showed a linear relationship without any ingredients ($R^2 > 0.85$). However, the deformation properties of SG-EW and SG-WP showed a non-linear relationship when the penetration distance was compared to failure ring tensile strain and shear strain. When the surimi gels were extremely compressed during the penetration test, the penetration distance was affected by the compressed gel matrix. This led to a converged penetration distance value at around 14 mm. Additionally, DIC analysis indicates that the penetration test induced a concentrated local strain at the fracture point of surimi gels regardless of the addition of ingredients. In contrast, the addition of EW and WP showed higher local strain concentration in surimi gels than those in surimi gels without any ingredients in the failure ring tensile test and torsion test. Therefore, this result indicates that the protein interaction of surimi gels with egg white and whey protein led to higher local concentrations in the torsion and ring tensile tests, while the penetration test could not detect that protein interaction in surimi gels. Local strain analysis using DIC can help in understanding the strain distribution of surimi gels and how it affects the overall deformation properties of surimi gels in penetration, tensile, and torsion tests.

**Author Contributions:** Conceptualization, H.W.P. and J.W.P.; formal analysis, H.W.P. and W.B.Y.; investigation, H.W.P. and W.B.Y.; writing—original draft preparation, H.W.P. and W.B.Y.; writing—review and editing, J.W.P., W.B.Y., and H.W.P.; supervision, J.W.P.; project administration, J.W.P. and H.W.P. All authors have read and agreed to the published version of the manuscript.

**Funding:** The following are results of a study on the "Leaders in Industry-university Cooperation 3.0" Project [202212050001], supported by the Ministry of Education and National Research Foundation of Korea.

**Institutional Review Board Statement:** Not applicable.

**Informed Consent Statement:** Not applicable.

**Data Availability Statement:** The data presented in this study are available on request from the corresponding author.

**Conflicts of Interest:** The authors declare no conflict of interest.

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
