# Peer review of "The Relationship between Penetration, Tension, and Torsion for the Fracture of Surimi Gels: Application of Digital Image Correlation (DIC)"

_processes, doi:10.3390/pr11010265_

Round 1

Reviewer 1 Report

Thank you for submitting your paper. I found this paper successful. The work done here draws attention to a significant subject. The experiments and sample preparation were well-designed and tested.

The abstract and conclusion are well explained.

The introduction is good and supported by scientific studies.

Material and method organization and descriptions are good. However, there are missing steps for digital image correlation. Please describe the lens properties of the camera. Please express the FoV of the imaging device. Please show the image processing algorithms used in MATLAB. Please add the unit conversion steps from pixel to mm. Please indicate the measuring stage that the software has automated procedures or manual.

The result section is well-designed, and the references are adequate.

In conclusion, the paper is well-designed and tested. On my side, there is no additional revision is needed for the publication.

Author Response

Dear Reviewer,

The authors would like to thank the editor and reviewers for the constructive review comments which helped to further strengthen the manuscript. Specific review comments and author responses are summarized below. Thank you.

Answers to Reviewer 1

Thank you for the comment. A digital compact camera (COOLPIX S6100, Nikon Co., Tokyo, Japan) with a lens (NIKKOR 5-35 mm f/3.7-5.6 25° FOV, Canon Inc,. Tokyo, Japan) was used in this study. After getting successive images from the videos, threshold-based segmentation was carried out to extract the target objects (surimi gels) from the background. Image acquisition and image processing are automated. However, the images were transferred from the camera to the computer manually. In this study, we calculated the local strain but not the distance. Thus, we did not need the unit conversion step from pixel to mm. The authors have revised the image processing part based on the reviewer’s comment. (L188-203)

Revised)

2.4.1. Image acquisition

The image acquisition systems for the penetration test, failure ring tensile test, and torsion test were developed as shown in Figure 1. A digital compact camera (COOLPIX S6100, Nikon Co., Tokyo, Japan) with a lens (NIKKOR 5-35 mm f/3.7-5.6 25° FOV, Nikon Co., Tokyo, Japan) was used to record each test at frame rates of up to 30 fps with a resolution of 0.95 million pixels during each test. For the penetration test, the camera was located to record the penetration point of the surimi gel at an angle of 15° (Figure 1a). To record the front view of the sample in the failure ring tensile test, the camera was placed horizontally over the sample at a distance of 140 mm (Figure 1b). For the torsion test, the camera was placed horizontally over the sample to focus on the center of the bottom part at a distance of 70 mm (Figure 1c). Images were obtained in PNG format from the video frames at each time interval. The obtained successive images were transferred to the computer for automated image processing, followed by DIC analysis. The white objects (surimi gels) placed on a contrasting background in the successive images were extracted by a thresh-old-based segmentation algorithm [5]. The processed images were used for DIC analysis.

Reviewer 2 Report

The textural properties of surimi gels made with or without whey protein concentrate, potato starch, and dried egg white were measured under torsion, tensile, and penetration tests. The correlation between the textural properties and deformation properties were investigated. Additionally, the local strain distributions of surimi gels were analyzed by DIC. Some interesting results have been provided. The reviewer thinks that the manuscript can be accepted by Processes, but there are some corrections required to improve the current form of the manuscript for publication.

1. In section 2.2, the addition concentration of EW, WP and PS should be indicated.

2. The statistical analysis methods should be provided in the Materials and Methods section.

3. The unit of textureal properties data in Table 1 should be provided.

4. What is the difference between SG and AP? In some section the SG and SG-EW, SG-WP and SG-PS were used, while the AP, AP-EW, AP-WP and AP-PS were used in other section. The explanation is needed.

5. The relationship between the textural properties and deformation properties of surimi gels were investigated and the DIC were used to analyzed the local strain of surimi gels. However, the correlation between the textural properties of surimi gels and DIC results could be discussed and summarized deeply.

Author Response

Dear Reviewer,

The author would like to thank the editor and reviewers for the constructive review comments which helped to further strengthen the manuscript. Specific review comments and author responses are summarized below. Thank you.

Reviewer #3

The textural properties of surimi gels made with or without whey protein concentrate, potato starch, and dried egg white were measured under torsion, tensile, and penetration tests. The correlation between the textural properties and deformation properties were investigated. Additionally, the local strain distributions of surimi gels were analyzed by DIC. Some interesting results have been provided. The reviewer thinks that the manuscript can be accepted by Processes, but there are some corrections required to improve the current form of the manuscript for publication.

C1. In section 2.2, the addition concentration of EW, WP and PS should be indicated.

R1. In this study, 250 batches of surimi gel were made to investigate the trends and relationships between different texture properties. In order to make 250 bathes of surimi gel, surimi gels were made at different moisture contents (74-81%) and different each ingredient level (0-9%). (L98-101)

2.2. Surimi gel preparation

Surimi paste and gel were prepared according to the method of Park et al. [5]. After thawing at 25 ºC for 1 h, surimi was cut into 5 cm cubes. A Stephan vacuum cutter UM-5 Stephan Machinery Corp., Columbus, OH, USA) was used to make surimi paste. In the first 1 min, frozen cubes were chopped at low speed. Salt (20 g/kg) was sprinkled in and chopping was continued at low speed for 1 min. Ice/water (0 ºC) with or without ingredients (0 - 90 g/kg), such as dried egg whites (SG-EW), whey protein concentrate (SG-WP), and potato starch (SG-PS), was added to adjust the moisture level to 740 - 810 g/kg, and the samples were chopped at low speed for 1 min.

C2. The statistical analysis methods should be provided in the Materials and Methods section.

R2. Thank you for the comment. The experiments were carried out in triplicated, and one-way Anova and Tukey’s multiple comparison test were conducted at the significance level of 0.05. Authors added the statistical analysis as suggested. (L218-221)

Revised)

2.5. Statistical analysis

All experiments were conducted 10 times. Mean values and standard deviations were determined. One-way ANOVA and Tukey’s multiple comparison tests were carried out at the significance level of 0.05 by using SPSS software (SPSS Inc, Chicago, IL).

C3. The unit of textural properties data in Table 1 should be provided.

R3. Among different texture properties in Table 1, only the penetration distance has a unit (mm).  In contrast to the penetration distance, failure ring tensile strain and shear strain are dimensionless. Thus, these two properties have no unit. (L111-186)

Texture analysis

2.3.1. Penetration test

Penetration tests were performed with a TA-XT texture analyzer (Stable Micro Systems, Surrey, UK) equipped with a spherical plunger (1 mm/sec of crosshead speed and 5 mm of diameter). Surimi gels at 4 ºC were placed at 25 ºC for 3 h prior to gel testing. Cylinder-shaped surimi gels (25 mm long) were prepared and subjected to fracture by penetration. The penetration distance and breaking force were measured in mm and g, respectively. All tests were conducted 10 times.

2.3.2. Ring tensile test

A failure ring tensile test was performed using a TA-XT texture analyzer equipped with 2 pins (diameter 10 mm) according to the method of Park and Yoon [5]. The cylindrical gels were cut into disk shapes (diameter, 30 mm; length, 10 mm). The ring-shaped samples were prepared by perforating the gels with a ring cutter. The ring sample dimensions had a width of 10 mm, an inside diameter of 17 mm, and a thickness of 5 mm. The pin at the top of the equipment was moved up to fracture the ring-shaped samples by tension (constant speed of 1 mm/sec). All experiments were conducted 10 times.

      In the ring tensile test in this study, failure hoop stress and failure uniaxial tensile stress were estimated to determine gel hardness (failure ring tensile stress). Failure hoop stress relates the wall circumferential stress to the internal pressure and the geometry of the ring specimen. Failure hoop stress was calculated by the following equation [5]:

                                                                                                                                           (1)

where  is the failure hoop stress,  is the load measured during ring tensile testing,  is the width of the ring specimen, and  is the inside diameter of the ring specimen. In failure uniaxial tensile stress, the sample is subjected to tension by opposing forces along its axis. Failure uniaxial tensile stress is calculated based on the cross-sectional area of the specimen as follows:

                                                                                                                                         (2) where  is the failure uniaxial tensile stress and  is the thickness of the ring specimen.

      In ring tensile testing, the ring specimen is deformed in the circumferential direction. Failure ring tensile strain was estimated according to Dieter [21], based on the dimensionless true strain by using the following equation:

                                                                                                                                          (3) where  is the failure ring tensile strain,  is the inside circumference of ring specimen, and  is the initial inside circumference of the ring specimen.

2.3.3. Torsion test

A torsion test was performed with a torsion gelometer (Gel Consultants, Raleigh, NC, USA). Cold gels (4 ºC) were placed at room temperature for 2 h to equilibrate gel temperatures before testing. Dumbbell-shaped samples were measured in the gelometer for torsional shear at a rotational rate of 2.5 rpm. All experiments were conducted 10 times.

      Shear stress and shear strain at mechanical failure were measured. Shear stress and shear strain indicate gel hardness and deformability of the gel, respectively [22-23]. Failure shear stress occurs at the boundary of the minimum cross-section for this specimen geometry. The shear stress value was calculated by the following equation [25-26]:

                                                                                                                        (4)

where  is the shear stress,  is the torque,  is the radius of the smallest cross-section of the torsion specimen,  is the shape factor constant, and  is the polar moment of inertia of the smallest cross-section.

      The shear strain is defined as follows [24]:

                                                                                     (5)

                                                                                     (6)

                                                                                                                                      (7)

where  is the total angular rotation,  is the angular rotation in the Q section,  is the angular rotation in the U section,  is the specimen shape factor of the Q section,  is the specimen shape factor of the U section,  is the shear modulus, and  is the dimensionless shear strain (Figure 1). Shear strain at the failure point was evaluated based on the true strain defined according to Nadai [25] using the following equation:

                                                                                      (8)

It is based on the material is elastic. Since the shear strain at failure is desired, if a material does not behave elastically up to the point of failure, difficulty arises in obtaining the actual failure shear strain value [21].

C4. What is the difference between SG and AP? In some section the SG and SG-EW, SG-WP and SG-PS were used, while the AP, AP-EW, AP-WP and AP-PS were used in other section. The explanation is needed.

R4. In Fig. 4 and Table 1, Alaska pollock surimi (A grade) (AP) was specifically chosen as a representative surimi gel to evaluate the local strain for each test method. After that, the local strain concentration of total 250 batches of surimi gels (SG) were investigated to see the trends (Fig. 5). From Fig. 5, we figured out that the different local strain trends between different test methods. Authors clarified that in the manuscript as shown below. (L277-278)

3.3. Evaluation of local strain

Based on the relationship between textural properties related to deformation (Figure 3), Alaska pollock surimi gels (A grade), which showed a similar penetration range, were chosen as representative surimi gels to compare the strain properties (Table 1). Surimi gels were made without (AP) and with 3% dried egg white, whey protein concentrate, or potato starch (AP-EW, AP-WP, and AP-PS, respectively). Failure ring tensile strain and shear strain of different surimi gels with or without added ingredients were significantly different in contrast to the penetration distance results, which showed similar value ranges. The strain values of AP-EW and AP-WP were lower than that of AP in the failure ring tensile test and torsion test. These differences related to local strain could be explained by the different effects of the ingredients in the formation of the network structure of the surimi gels [1,35-37,40].

C5. The relationship between the textural properties and deformation properties of surimi gels were investigated and the DIC were used to analyze the local strain of surimi gels. However, the correlation between the textural properties of surimi gels and DIC results could be discussed and summarized deeply.

R5. Thank you for the comment. Authors revised the manuscript to discuss how the local strain properties analyzed by DIC affected the correlation between the different textural properties of surimi gels. (L350-358)

Revised)

    Conclusions

This study investigated the fracture properties of surimi gels with and without added ingredients. The textural properties related to the hardness and the deformation of surimi gels were measured by penetration, ring tensile, and torsion tests, which showed a linear relationship without any ingredients (R2 > 0.85). However, the deformation properties of SG-EW and SG-WP showed a non-linear relationship when the penetration distance was compared to failure ring tensile strain and shear strain. When the surimi gels were extremely compressed during the penetration test, the penetration distance was affected by the compressed gel matrix. It led to a converged penetration distance value of around 14 mm. Additionally, DIC analysis indicates that the penetration test induced a concentrated local strain at the fracture point of surimi gels regardless of the addition of ingredients, whereas the addition of EW and WP showed higher local strain concentration in surimi gels than those in surimi gels without any ingredients in the failure ring tensile test and torsion test. Therefore, this result indicates that the protein interaction of surimi gels with egg white and whey protein led to higher local concentrations in the torsion and ring tensile tests, while penetration test could not detect that protein interaction in surimi gels. Local strain analysis using DIC can help in understanding the strain distribution of surimi gels and how it affects the overall deformation properties of surimi gels in penetration, tensile, and torsion tests.

Reviewer 3 Report

Dear Authors,

Here are the reviewer's comments: 1. The abstract is too short, there are no hypotheses or research questions. 2. Figures 2 and 3 are hard to read. They should be cut into smaller pieces and/or arranged differently. In Figure 5, explain in the legend below the figure what the letters a, b mean. 3. The conclusions should be slightly expanded and numbered specifically 1,2,3, etc. Literature items should not be older than 2000. Replace older items with newer ones both in the bibliography and in the text, e.g. items 14, 20-27, 33, 35-36, 47, 49. After taking into account the above corrections, the publication can be published.

Regards

Author Response

Dear Reviewer,

The authors would like to thank the editor and reviewers for the constructive review comments which helped to further strengthen the manuscript. Specific review comments and author responses are summarized below. Thank you.

C1. The abstract is too short, there are no hypotheses or research questions.

R1. Authors have revised the abstract to address the research questions we had. (L12-15)

Revised)

Abstract

A standardized method to evaluate the material properties of surimi gels has to be updated because of the lack of accuracy and the repeatability of data obtained from conventional ways.  To investigate the relationships between the different texture measurement methods used in surimi gels, 250 batches of different surimi gels were used. The textural properties of surimi gels made with or without whey protein concentrate (SG-WP), potato starch (SG-PS), or dried egg white (SG-EW) were measured under torsion, tensile, and penetration tests. The correlation between the textural properties related to the deformation and hardness of surimi gels without any added ingredients (SG) was linear (R2 > 0.85). However, the R2 values of the shear strain and tensile strain of SG-WP and SG-EW were significantly lower than that of SG. The strain distributions of surimi gels with and without added ingredients were estimated by digital image correlation (DIC) analysis. The results showed that the local strain concentration in SG-WP and SG-EW was significantly higher than that of SG in the failure ring tensile test and the torsion test (P < 0.05). DIC analysis was an effective tool for evaluating the strain distribution characteristics of surimi gels upon fracture from torsion, penetration, and tension.

C2. Figures 2 and 3 are hard to read. They should be cut into smaller pieces and/or arranged differently. In Figure 5, explain in the legend below the figure what the letters a, b mean.

R2. For better visibility, the font size of captions and legends in figures 2 and 3 have been increased. The letters a and b in each test method mean that values are significantly different (p < 0.05). Authors described the meaning of different letters in Fig 5 as well as Table 1.

Revised)

Figure 2.

Figure 3.

C3. The conclusions should be slightly expanded and numbered specifically 1,2,3, etc.

R3. Authors revised the conclusion part to add more discussion on how the local strain properties analyzed by DIC affected the correlation between the different textural properties of surimi gels. (L350-358)

Revised)

    Conclusions

This study investigated the fracture properties of surimi gels with and without added ingredients. The textural properties related to the hardness and the deformation of surimi gels were measured by penetration, ring tensile, and torsion tests, which showed a linear relationship without any ingredients (R2 > 0.85). However, the deformation properties of SG-EW and SG-WP showed a non-linear relationship when the penetration distance was compared to failure ring tensile strain and shear strain. When the surimi gels were extremely compressed during the penetration test, the penetration distance was affected by the compressed gel matrix. It led to a converged penetration distance value at around 14 mm. Additionally, DIC analysis indicates that the penetration test induced a concentrated local strain at the fracture point of surimi gels regardless of the addition of ingredients. In contrast, the addition of EW and WP showed higher local strain concentration in surimi gels than those in surimi gels without any ingredients in the failure ring tensile test and torsion test. Therefore, this result indicates that the protein interaction of surimi gels with egg white and whey protein led to higher local concentrations in the torsion and ring tensile tests, while the penetration test could not detect that protein interaction in surimi gels. Local strain analysis using DIC can help in understanding the strain distribution of surimi gels and how it affects the overall deformation properties of surimi gels in penetration, tensile, and torsion tests.

C4. Literature items should not be older than 2000. Replace older items with newer ones both in the bibliography and in the text, e.g. items 14, 20-27, 33, 35-36, 47, 49.

R4. Besides references describing the theoretical backgrounds, old references has been replaced by new references as suggested by the reviewer.

Removed)

Weerasinghe, V.C.; Morrissey, M.T.; Chung, Y.; An, H. Whey protein concentrate as a proteinase inhibitor in Pacific Whiting surimi. Journal of Food Science 1996, 61(2), 367-371.

Yoon, W.B.; Park, J.W.; Kim, B.Y. Linear programming in blending various components of surimi seafood. Journal of Food Science 1997, 62(3), 561-564.

Tang, J.; Tung, M.A.; Lelievre, J.; Zeng, Y. Stress-strain relationships for gellan gels in tension, compression and torsion. Journal of Food Engineering 1997, 31, 511-529.

Lee, C.M.; Kim, J.M. The relationship of composite characteristics to rheological properties of surimi gel. In Food Engineering and Process Applications; Maguer, M.L., Jelen, P., Eds.; Elsevier Applied Science: London, UK, 1986.

Lee, C.M.; Wu, M.C.; Okada, M. Ingredient and formulation technology for surimi-based products. In Surimi Technology; Lanier, T.C., Lee, C.M., Eds.; Marcel Dekker, Inc.: New York, USA, 1992; pp 273-302.

Yang, H.; Park, J.W. Effects of starch properties and thermal-processing conditions on surimi-starch gels. LWT-Food Science and Technology 1998, 31(4), 344-353.

Added)

Tabilo-Munizaga, G.; Barbosa-Canovas G.V. Color and textural parameters of pressurized and heat-treated surimi gels as affected by potato starch and egg white. Food Research International 2004, 37(8), 767-775.

Hunt, A.; Getty, K.J.K.; Park, J.W. Roles of starch in surimi seafood: a review. Food Reviews International 2009, 25(4), 299-312.

Li, X.; Fan, M.; Huang, Q.; Zhao, S.; Xiong, S.; Yin, T.; Zhang, B. Effect of micro- and nano-starch on the gel properties, mi-crostructure and water mobility of myofibrillar protein from grass carp. Food Chemistry 2022, 366, 130579.

Round 2

Reviewer 2 Report

I would like to thank the authors for considering my comments and answering my questions. After careful reading of the answers to my questions I can recommend accepting the manuscript by Processes.